# Otilonium Bromide Prevents Cholinergic Changes in the Distal Colon Induced by Chronic Water Avoidance Stress, a Rat Model of Irritable Bowel Syndrome

**DOI:** 10.3390/ijms24087440

**Published:** 2023-04-18

**Authors:** Chiara Traini, Eglantina Idrizaj, Cristina Biagioni, Maria Caterina Baccari, Maria Giuliana Vannucchi

**Affiliations:** 1Histology and Embryology Research Unit, Department of Experimental and Clinical Medicine, University of Florence, 50139 Florence, Italy; chiara.traini@unifi.it (C.T.);; 2Section of Physiological Sciences, Department of Experimental and Clinical Medicine, University of Florence, 50139 Florence, Italy

**Keywords:** mucus secretion, choline acetyl transferase, muscarinic receptors, protein gene product 9.5, tetrodotoxin, atropine, methacholine, corticotropin-releasing factor, contractile responses

## Abstract

Irritable Bowel syndrome (IBS) is a highly widespread gastrointestinal disorder whose symptomatology mainly affect the large intestine. Among the risk factors, psychosocial stress is the most acknowledged. The repeated water avoidance stress (rWAS) is considered an animal model of psychosocial stress that is capable of mimicking IBS. Otilonium bromide (OB), which is orally administered, concentrates in the large bowel and controls most of the IBS symptoms in humans. Several reports have shown that OB has multiple mechanisms of action and cellular targets. We investigated whether the application of rWAS to rats induced morphological and functional alterations of the cholinergic neurotransmission in the distal colon and whether OB prevented them. The results demonstrated that rWAS affects cholinergic neurotransmission by causing an increase in acid mucin secretion, in the amplitude of electrically evoked contractile responses, abolished by atropine, and in the number of myenteric neurons expressing choline acetyltransferase. OB counteracted these changes and also showed an intrinsic antimuscarinic effect on the post-synaptic muscular receptors. We assume that the rWAS consequences on the cholinergic system are linked to corticotrophin-releasing factor-1 (CRF1) receptor activation by the CRF hypothalamic hormone. OB, by interfering with the CFR/CRFr activation, interrupted the cascade events responsible for the changes affecting the rWAS rat colon.

## 1. Introduction

Irritable bowel syndrome (IBS) is one of the most common gut disorders, with prevalence rates between 5 and 10% in Western countries and China [1]. The symptoms and signs, which are mainly related to the large intestine, consist of visceral pain and/or discomfort, hypersensitivity, abnormal motor responses, and changes in bowel habits [2]. Clinically, IBS is a chronic disease characterized by the fluctuation of remission periods and periods of exacerbation symptoms [2].

The etiopathogenesis of IBS is still unclear. Notably, up to 80% of IBS patients experience comorbid behavioral disorders, such as anxiety or depression [3,4]. Furthermore, psychosocial stress events are widely recognized [5,6] as risk factors for the development and/or relapse of IBS symptoms, probably via the gut–brain axis [7,8,9,10]. Finally, although IBS is commonly defined as a functional disorder, the presence of mucosal barrier impairment, low-grade inflammation, gut–brain axis dysfunction, and dysbiosis suggests the existence of an organic component [11,12].

Due to the difficulty in obtaining human samples and the importance of psychosocial stressors in IBS, some animal models of stress have been developed in an attempt to mimic the symptomatology of the disease [5]. Among them, chronically applied water avoidance stress (WAS), i.e., repeated (r)WAS, is considered one of the most effective approaches [5,7,13,14]. Applied to the correct rat strain, the Wistar, rWAS induces two different types of stress: isolation, and immobility, and after ten days of repeated exposure, it causes the development of chronic enteric hyperalgesia, increased production of fecal pellets, and a mild mucosa inflammation [7,13,14]. Recently, we also demonstrated that 10 days of rWAS caused immobility when the EPM behavioral was tested, which abolished the giant contractions in the circular muscle layer and increased the iNOS expression in the myenteric neurons of the distal colon [15]. Much less is known about the potential role of the cholinergic system in the genesis of IBS. The few available studies are limited to the role of this neuronal pathway in the development of mucosal inflammation [16,17].

Otilonium bromide (OB), a quaternary ammonium derivative, is a global and effective drug used to control the signs and symptoms of IBS. When administered orally, it is poorly absorbed and exerts its activity mainly in the large intestine [18,19]. It acts on different cell types, i.e., smooth muscle cells and neurons, with multiple mechanisms of action: it antagonizes the tachykinin and muscarinic receptor activation, binds to L-type calcium channels, interferes with neurotransmission, and affects the bacterial population of the microbiota [15,20,21,22,23,24]. Thus, understanding the cellular mechanisms that are responsible for the efficacy of OB in controlling IBS symptoms could help in comprehending the pathogenesis of the disease.

Starting from this information, we designed a study to test whether rWAS rats develop colonic neuro-muscular abnormalities imputable to the cholinergic pathway and whether OB treatment could prevent them. These objectives were achieved by combining functional and morphological studies.

## 2. Results

### 2.1. Body Weight Gain, Water Intake, and Fecal Pellet Production

All the animals were weighed before the beginning of the rWAS application, every two days during the stress period, and at the end of the treatment. No significant differences were observed among the experimental groups at any time point (Appendix A). The water intake, which was evaluated daily, was not different among the Ctrl, Sham, and rWAS groups (Ctrl = 42.34 ± 1.5 mL; Sham = 40.50 ± 2.2 mL; and rWAS = 43.38 ± 2.1); while, as expected, the bitter taste of OB dissolved in the drinking water caused a significant decrease in the water intake in OB and rWAS+OB groups (OB = 31.42 ± 1.2 mL; rWAS+OB = 30.59 ± 1.6 mL); (*p* < 0.005). The mean number and mean weight of the fecal pellets were significantly higher in the rWAS and rWAS+OB groups compared to the Sham group (Appendix A, gray columns). With time, the fecal pellet production decreased significantly in the Sham and rWAS+OB groups (Appendix A, blue columns) suggesting the appearance of environment habituation for the former and confirming the drug efficacy for the latter. Conversely, in the rWAS rats, the fecal production did not decrease with time (Appendix A, blue columns). The fecal production in controls and OB rats during the ten experimental days did not differ from that obtained in the Sham [15]).

### 2.2. Histochemistry

#### Periodic Acid and Schiff’s Reagent (PAS) and Toluidine Blue Staining

The PAS staining (Figure 1A–C), which estimates the mucus production, was similar among the groups of animals. In the OB group, the secretion was the highest. Quantitation of the staining confirmed these observations, but the result obtained in OB rats did not reach the statistical significance (Figure 1G, right columns). The Toluidine Blue staining (Figure 1D–F), which estimates the quality of the mucus (i.e., the acidity), was significantly increased in rWAS rats compared to the other groups of animals (Figure 1G, left columns). 

### 2.3. Functional Studies

#### 2.3.1. Contractile Responses Elicited by Electrical Field Stimulation (EFS) in Strips from the Different Animal Groups

In preparations from Ctrl rats, EFS (4 Hz) induced a rapidly arising contractile response (mean amplitude 0.97 ± 0.19 g) (Figure 2A–C), which was abolished by TTX or atropine, indicating its nervous and cholinergic nature, respectively (Figure 2A). In a minority (20%) of recordings, no responses were elicited during EFS but at the end of the stimulation period, an “after contraction” was observed. These responses did not enter the statistics. The amplitude of the contractile responses to EFS in preparations from Ctrl rats was not statistically different from that obtained in Sham rats (mean amplitude 0.95 ± 0.16 g). Thus, the statistical data were put together and analyzed (Figure 2C). 

In preparations from rWAS rats, the amplitude of the contractile responses elicited by EFS (mean amplitude 1.6 ± 0.20 g) was greatly enhanced with respect to that obtained from the Ctrl/Sham animals (Figure 2B,C). In strips from OB animals, the amplitude of the EFS-induced contractile responses (mean amplitude 0.52 ± 0.13 g) was reduced with respect to the Ctrl/Sham, and even more with respect to the rWAS groups (Figure 2B,C). In preparations from rWAS+OB rats, the amplitude of the EFS-induced contractile response (mean amplitude 0.63 ± 0.12 g) was reduced with respect to either the Ctrl/Sham or rWAS groups, but it was not statistically different from that obtained in OB animals (Figure 2B,C). 

As in the Ctrl preparations, the EFS contractile responses from all the other animal groups were abolished by TTX or atropine.

#### 2.3.2. Direct Smooth Muscle Contractions Elicited by Methacholine in Strips from the Different Animal Groups

To investigate whether the reduction in the amplitude of the EFS-induced contractile responses observed in OB and rWAS+OB preparations could be due to a direct effect of OB on the smooth muscle muscarinic receptors, the effects of methacholine were tested and compared to those obtained in the other animal groups. As previously observed [15], in preparations from both Ctrl and rWAS rats, the addition of methacholine to the bath medium caused, after 10–15 s of contact time, a sustained contracture, which reached a plateau phase that persisted until washout (Figure 3A). In the present experiments, the amplitude of the contractile responses to methacholine in preparations from Ctrl rats (n = 6) was not statistically different from that obtained in Sham rats (mean amplitude 1.08 ± 0.1 g and 0.99 ± 0.15 g, respectively). Thus, the statistical data were put together and together (Figure 3B). No statistical differences were observed in the amplitude of the direct muscular response to methacholine between the Ctrl/Sham animal group and rWAS rats (mean amplitude 1.04 ± 0.1 g and 1 ± 0.16 g, respectively) (Figure 3B). The amplitude of the direct muscular responses to methacholine obtained in preparations from either OB (mean amplitude 0.57 ± 0.11 g) or rWAS+OB (mean amplitude 0.64 ± 0.08 g) rats was reduced in comparison with that of both Ctrl/Sham and rWAS animal groups, thus indicating that the observed reduction in amplitude of the EFS-induced cholinergic contraction in OB and rWAS+OB was, at least in part, ascribable to a direct antimuscarinic effect of OB on the smooth muscle. 

### 2.4. Immunohistochemistry

#### 2.4.1. Protein Gene Product 9.5 (PGR9.5) Immunoreactivity (IR)

In all the groups of animals, the pan-neuronal marker PGP9.5-IR was detected in the neurons of both enteric plexuses and in the nerve fibers of the muscle coat (Figure 4A–D). The labeling was homogenously distributed in the cytoplasm. The quantitation of the myenteric neurons expressing PGP9.5 displayed no significant differences among the groups (Ctrl = 80 ± 5.2; Sham = 79.47 ± 4.9; rWAS = 78.81 ± 3.1; OB = 85.00 ± 3.2; and rWAS+OB = 80.41 ± 3.5). 

#### 2.4.2. Choline Acetyl Transferase (ChAT)-IR

ChAT-IR was observed in the cytoplasm of several myenteric neurons and in few intramuscular nerve fibers (Figure 4E–H); the labeling was homogenously distributed. The quantitation of the ChAT-IR neurons, expressed as total numbers as well as percentage of PGP9.5-IR neurons, showed a significant increase in the rWAS rats in comparison to all the other groups (Figure 4I). 

## 3. Discussion

The present study demonstrates that daily exposure to one hour of psychosocial stress for 10 days induces relevant changes in cholinergic neurotransmission in the distal colon of rats. In particular, rWAS causes the following: (i) a significant increase in the acidic component in the epithelium without changes in the total mucus secretion; (ii) a significant increase in the amplitude of the contractile responses to electrical stimulation, which was impeded by TTX and atropine; and (iii) an increase in the number of ChAT-IR myenteric neurons. OB treatment prevented most of these changes. 

In a previous study [15], we showed that the rWAS procedure fits with the criteria proposed by Meyer and Collins [25] to validate an animal model and that the colonic alterations observed were superimposable to those observed in the IBS patients [5,13,15]. In particular, the increased fecal production in rWAS rats during the stress application is a clear sign of colonic hyperactivity [9,10,15].

In the present work, we report a significant increment of the acidic mucus secretion in rWAS rats. The mucus covers the entire gastrointestinal surface and plays a critical role in local homeostasis in terms of conditioning the bacterial population and the luminal pH [26,27]. The mucus quality has never been investigated in IBS patients; nevertheless, these patients show dysbiosis in more than 70% of cases [28,29]. Dysbiosis alters the intestinal barrier, causes low-grade mucosal inflammation, and changes the mucus quality [28,29]. In turn, changes in mucus composition are responsible for dysbiosis [26,27], thus creating a vicious circle. Based on the literature data showing that acetylcholine regulates the epithelial ion transport, the colonic crypt activity [30,31], and the mucus secretion [32], we hypothesized that the activation of the cholinergic pathways (i.e., those located at the submucous plexus) could be responsible for the modified mucosal secretion. If so, the ability of OB to prevent changes in mucus secretion in rWAS+OB rats could be due to its antimuscarinic activity. In fact, it has been reported that OB controls submucosal cholinergic activation by inhibiting the M3 receptors [30]. Furthermore, as it has been documented, a bactericidal property of OB against pathogens [24] the drug could also indirectly prevent mucus acidification through this activity on the microbiota.

In the functional experiments, the increased amplitude of electrically induced contractile responses in rWAS preparations compared to the Ctrl/Sham ones and the similar amplitude of the direct muscular response to methacholine between the two animal groups indicate that the rWAS animal model selectively involves the cholinergic output without affecting the post-synaptic muscular response. The observation that, in all animal groups, the EFS-induced contractile responses were abolished by TTX or atropine confirms their nervous and cholinergic origins, respectively. The increase in ChAT-IR myenteric neurons in the rWAS rats is consistent with the main role of the cholinergic system. The increased amplitude of the neurally induced contractions elicited by EFS observed in preparations from rWAS was counteracted by OB treatment (rWAS+OB). However, the effect of OB on the colonic mechanical responses was more complex than expected. In fact, in the rWAS+OB preparations and in those of rats treated only with OB, the amplitude of neurally induced contractile responses to EFS was significantly lower than that of Ctrl/Sham, suggesting the presence of an intrinsic antimuscarinic action of the drug [23], regardless of the application of stress. In favor of a direct antimuscarinic effect of OB is the observation that the amplitude of the muscular responses to methacholine was reduced in both OB and rWAS+OB preparations in comparison with Ctrl/Sham and rWAS rats. Thus, the functional results suggest that OB decreases the amplitude of EFS-induced contractile responses at least in part through an antimuscarinic activity. Whether this decrease in the EFS in rWAS+OB preparations comprises both, presynaptic and post-synaptic cholinergic structures cannot be inferred from these studies, as the neuronal effect could be masked by the direct antimuscarinic post-synaptic effects of the drug on smooth muscle. 

Immunohistochemical experiments showed that stress activates the cholinergic system with a significant increase in the number of ChAT-IR in rWAS. This increase was prevented by OB. Thus, combining the immunohistochemical and functional data, it can be deduced that the effect of OB on the cholinergic transmission rests on a dual mechanism of action: on the neurons (presynaptic) and on the smooth muscle cells (post-synaptic). 

The hypothalamic corticotrophin-releasing hormone (CRF) is considered the main mediator of the psychosocial stress in mammals [33,34,35,36]. Centrally or peripherally injected, CRF causes adverse intestinal responses that mimic those produced by stress [33,37,38]. These responses are due to CRF1 and CRF2 receptor activation. Precisely, the CRF1r mediates the adverse effects, while the CRF2r, which ‘per se*’* has no effects, hinders CRF1r activation. The hormone binds with a higher affinity to the CRF1r [33,36,39,40]. CRF1r is present in almost 95% of the myenteric neurons in the rat’s distal colon, and when it binds to CRF, 50% of these neurons show a significant increase in *c-fos* [33,39,41]. Co-labeling experiments revealed that *c-fos* was expressed in 40% of the cholinergic neurons [39] and that its activation promoted the transcription of neurotransmitter biosynthetic enzymes, including choline acetyltransferase [42]. Functionally, *c-fos* activation increased defecation [39]. Thus, it can be inferred that, in rWAS rats, CRF1r-*c-fos* activation causes ChAT gene transcription and, in turn, a ChAT-IR myenteric neuron increase.

As described above, OB treatment significantly counteracts the effects of chronic stress on the cholinergic system. We previously reported that the chronic administration of OB in rats subjected to a different psychosocial stress prevented the increase in the CRF1r in the colonic myenteric neurons without affecting the simultaneous increase in the CRF2r [22,23]. Furthermore, it was shown in a rat colitis model that OB significantly reduced *c-fos* expression in lumbosacral spinal cord neurons [43]. 

## 4. Materials and Methods

### 4.1. Animals

Male Wistar rats (n = 30) weighing 150–200 g were purchased from Charles River Laboratories Italia srl (Lecco, Italy). The animals were housed 2–3 per cage at CeSAL (Department of NEUROFARBA, University of Florence, Florence, Italy) with a 12/12 h light/dark cycle at 22 °C room temperature (RT), and with free access to food and water. The experimental procedures were carried out in accordance with European guidelines for the care and use of laboratory animals (Directive 2010/63/UE) and were approved by the Italian Ministry of Health (code: 916/16). Efforts were conducted to maintain animal’s health, safety, and welfare to minimize suffering and to reduce the number of animals used in the experiments. On the day of arrival, the rats were transferred into the room adjacent to the testing room and stayed there for the entire period of the experiment. The rats were randomly divided into five experimental groups: (1)Group of control rats who were not exposed to stress or to pharmacological treatment (Ctrl; n = 4);(2)Group of rats who were subjected to repeated water avoidance stress (rWAS; n = 7) for 10 consecutive days;(3)Group of rats who experienced the stress environment without being subjected to it (Sham; n = 5);(4)Group of rats who were subjected to rWAS for 10 consecutive days and, meanwhile, they were orally treated with OB (rWAS+OB; n = 7);(5)Group of rats who were treated orally with OB for 10 days and not subjected to stress (OB; n = 7).

Each rat was weighed every 2 days to assess their weight gain.

### 4.2. Repeated Water Avoidance Stress (rWAS) Protocol

The rWAS was applied as reported in Traini et al. [15]. Briefly, a Plexiglas tank (45 cm length, 25 cm width, and 25 cm height) with a polygonal platform (10 cm length, 8 cm width, and 8 cm height) was fixed to the center of the tank floor and filled with fresh water (25 °C) up to 1 cm above the top of the platform. The water was dyed using a non-toxic dark color and it was changed before each section to remove the smell of the previous rat and to collect the fecal pellets. The animals were placed on the platform for 1 h/day for 10 consecutive days between 9:00 am and 02:00 pm, in accordance with their circadian rhythm. Sham rats were placed on the platform in the waterless tank for 1 h daily for 10 consecutive days. The locomotor activity of the rats was recorded by a video-tracking/computer-digitizing system (HVS Image, Hampton, UK) and the time spent by the animal on the platform (zone 2) or swimming in the water (zone 1) was quantified and used to evaluate the effective application of immobility stress. At the end of each daily exposition, the fecal pellets collected in the tank were counted, stored in separated bins (one for each animal), and exposed to the same conditions of temperature and humidity for 24 h to allow dehydration. Then, the pellets were weighed. Increments in number and weight of the fecal pellet were considered parameters to evaluate the colonic hyperactivity.

### 4.3. OB Preparation and Administration

OB (10 mg/kg/day) was added to the drinking water the day before the start of the rWAS, and its concentration was adjusted every 2 days based on body weight gain and water intake. Intake of the drug with water was checked by measuring residual water in the trough every 2 days.

### 4.4. Tissue Sampling

The following day after the last rWAS application, the rats were anesthetized and killed. The abdomen was opened, and the colon was rapidly removed. The distal colon was divided in two segments: one for the functional experiments, and the other for the morphological and biomolecular studies. For a detailed description of the procedures, see the sections below.

### 4.5. Functional Experiments

As previously described [44], two full-thickness circular muscle strips (0.4 ± 0.1 cm) were dissected from each colon segment and mounted in 5 mL organ baths containing Krebs–Henseleit solution composed of (in mM) 118 NaCl, 4.7 KCl, 1.2 MgSO_4_, 1.2 KH_2_PO_4_, 25 NaHCO_3_, 2.5 CaCl_2_, and 10 glucose, pH 7.4, and bubbled with 95% O_2_ 5% CO_2_. Temperature was maintained within a range of 37 ± 0.5 °C. One end of each strip was tied to a platinum rod, while the other was connected to a force displacement transducer (FT03; Grass Instrument, Quincy, MA, USA) by a silk thread for continuous recording of isometric tension. The transducer was coupled to a polygraph (7K; Grass Instrument, Quincy, MA, USA).

Strips equilibrate for 1 h under an initial load of 1 g. During this period, the preparations underwent repeated and prolonged washes with Krebs–Henseleit solution to prevent accumulation of metabolites in the organ baths.

Electrical field stimulation (EFS) was applied via two platinum wire rings (2 mm in diameter, 5 mm in apart) through which the preparation was threaded. Electrical pulses (rectangular waves, 80 V, 4 Hz, 0.5 ms for 15 s) were provided by a Grass model S8 stimulator. 

Direct smooth muscle contractions were obtained by adding methacholine to the bath medium. The interval between two subsequent applications of methacholine was no less than 30 min, during which repeated and prolonged washes of the preparations with Krebs–Henseleit solution were performed. 

The following drugs were used: the nerve blocker tetrodotoxin (TTX, 1 × 10^−6^ M), the muscarinic receptor agonist methacholine (2 × 10^−6^ M), and the muscarinic receptor antagonist atropine (1 × 10^−6^ M). Drug concentrations were those previously used in rodent gastrointestinal preparations and proved to be effective [45]. All drugs were obtained from Sigma (St. Louis, MO, USA). Solutions were freshly prepared, except for TTX, for which a stock solution was kept at −20 °C.

### 4.6. Morphological Studies

The full-thickness segments of distal colon were fixed in 4% paraformaldehyde in 0.1 M phosphate-buffered saline (PBS, pH 7.4) overnight (ON) at 4 °C, dehydrated in graded ethanol series, cleared in xylene, and embedded in paraffin with the cut section transversal to the longitudinal axis. Sections of 5 µm-thick segments were cut using a rotary microtome (MR2, Boeckeler Instruments Inc., Tucson, AZ, USA) and collected on positively charged slides. All the paraffin-embedded sections were deparaffinized and rehydrated as usual for routine histology, histochemistry, and immunohistochemistry. Some sections were stained with hematoxylin–eosin (H&E) to evaluate the tissue integrity, and others were submerged for 10 min in Toluidine Blue 0.1% in 30% ethanol, after filtering, and others were treated for Periodic Acid and Schiff reagent (PAS) staining; see [46] for details. At the end of each staining procedure, the slides were dehydrated, clarified, and mounted in synthetic resin. For immunohistochemical experiments, the deparaffinized sections were treated for 20 min at 90–92 °C in tris buffer (10 mM) with EDTA (1 mM, pH 9.0), followed by cooling to RT for antigen retrieval. Then, they were washed in PBS and blocked for 20 min at RT with 1.5% bovine serum albumin (BSA, Applichem, Darmstad, Germany) in PBS. The number of total myenteric neurons and the number of cholinergic positive myenteric were estimated using the rabbit polyclonal PGP9.5 (1:200; Millipore Corporation, Temecula, CA, USA) pan-neuronal antibody and the goat polyclonal choline acetyl transferase (ChAT) (1:100; Millipore Corporation, Temecula, CA, USA) antibody, respectively. As both antibodies were polyclonal, sequential sections were collected on slides (four sections/slide, two slides/animal) in two separate areas, one area containing the first and third sections, the other area containing the second and fourth sections. Each area was bordered with a pap pen, and the two sections of one area were incubated with the pan-neuronal marker PGP9.5 and the two sections with the ChAT antibody. This procedure made it possible to recognize the same ganglia and count the same neurons. The primary antibodies that were diluted in 1.5% of BSA in PBS were applied ON at 4 °C. The day after, the sections were washed in PBS and incubated for 2 h at RT in the dark with appropriate fluorochrome-conjugated (Invitrogen, San Diego, CA, USA) secondary antibodies diluted in PBS. The sections were washed in PBS, incubated for 10 min with Hoechst 33342, a nuclei marker (20 μg/mL; Sigma), dissolved in PBS, washed in distilled water, and mounted in an aqueous medium (Immuno-Mount, Thermo Scientific, Rockford, IL, USA). Negative controls were simultaneously performed, omitting the primary antibody to exclude the presence of non-specific immunofluorescence staining. The reaction was observed under the Olympus BX63 fluorescence microscope (Olympus, Segrate, Italy), the signal was obtained using 488- and 370 nm excitation wavelengths for the green and blue, fluorescent labels, respectively, and the photos were taken via the associated imaging system (CellSens Dimension Imaging Software, Olympus, Italy).

### 4.7. Quantitative and Statistical Analysis

The animal weight and the parameters related to fecal pellet (number and weight) were quantified and expressed as means ± S.E.M. The number of animals used is indicated as n in the legends.

Amplitude of contractile responses is expressed as absolute values (grams). Responses to EFS referred to the maximal peak obtained during the stimulation period, whereas contractions to methacholine were measured 30 s after a stable plateau phase was reached. Results are given as means ± S.E.M. The number of muscle preparations is reported in the results. Digital images of PAS- and TB-stained structures were acquired with a video camera-equipped microscope (Eclipse 200; Nikon Instruments, Tokyo, Japan) with ×10 objective, and the reconstruction of the entire section (4 sections/slide, 2 slides/animal) was performed using appropriate software. Quantitations were performed with ImageJ software (NIH, Bethesda, ML, USA) using the color threshold toll and expressed as pixel^2^/section.

The count of the number of PGP9.5 and ChAT positive myenteric neurons in the entire transverse section (4 sections/slide; 2 slides/animal) was performed by two observers (CT; EI) who were blind to each other, along the entire section, using a 20× objective. Only the neurons containing the nucleus were included. The results were expressed as the total number of positive neurons and as percentage of ChAT out of the PGP9.5 positive neurons per section ± S.E.M. The number of animals used is reported as n in the legends.

Statistical analysis was performed by means of paired or unpaired Student’s *t*-test to compare two experimental groups or one-way ANOVA followed by Newman–Keuls post-test when more than two groups were compared. Values were considered significantly different with *p* < 0.05. 

## 5. Conclusions

Our investigations of the pathogenesis of IBS using an appropriate animal model (rWAS) and the treatment of OB led us to draw a composite picture where the combination of morphological and functional experiments proved to be essential. In the previous paper by Traini et al. [15], who used the same model and drug treatment, we showed that a depression of spontaneous contractility is associated with a significant increase in the inhibitory nitrergic neurons. In the present work, we detected an increase in the electrically evoked responses of the colonic wall of rWAS rats that associates with an Increase in the excitatory cholinergic neurons. All together, these data allow us to state that psychosocial stress affects both of the two main mediators of enteric motility. Moreover, starting from the well-known role of CRF as a mediator of psychosocial stress [33,34] and based on the available data demonstrating that CRF receptors are highly expressed in both nitrergic and cholinergic neurons [39,40], we consider that the hypothalamic hormone plays a major role in the pathogenesis of IBS. Thus, we propose that OB, by preventing CRF1r activation, is able to interrupt the cascade of events that cause the functional and immunohistochemical changes described in rWAS rats. What is noteworthy is that the reduced response to methacholine in OB-treated rats indicates that OB exerts ‘per se*’* a post-synaptic antimuscarinic effect, independently of stress, and confirms the complexity of its mechanism of action [23]. 

In this perspective, it will be interesting to evaluate blood and brain corticotropin levels during the application of stress with the aim of better understanding the mechanisms of action of OB [18,23,47,48]. Finally, from a clinical point of view, our results strengthen the hypothesis formulated in some clinical studies [49,50] that cyclical or intermittent treatment with OB, regardless of the presence of IBS symptoms, may be an appropriate therapeutic strategy to manage the disease.

## Figures and Tables

**Figure 1 ijms-24-07440-f001:**
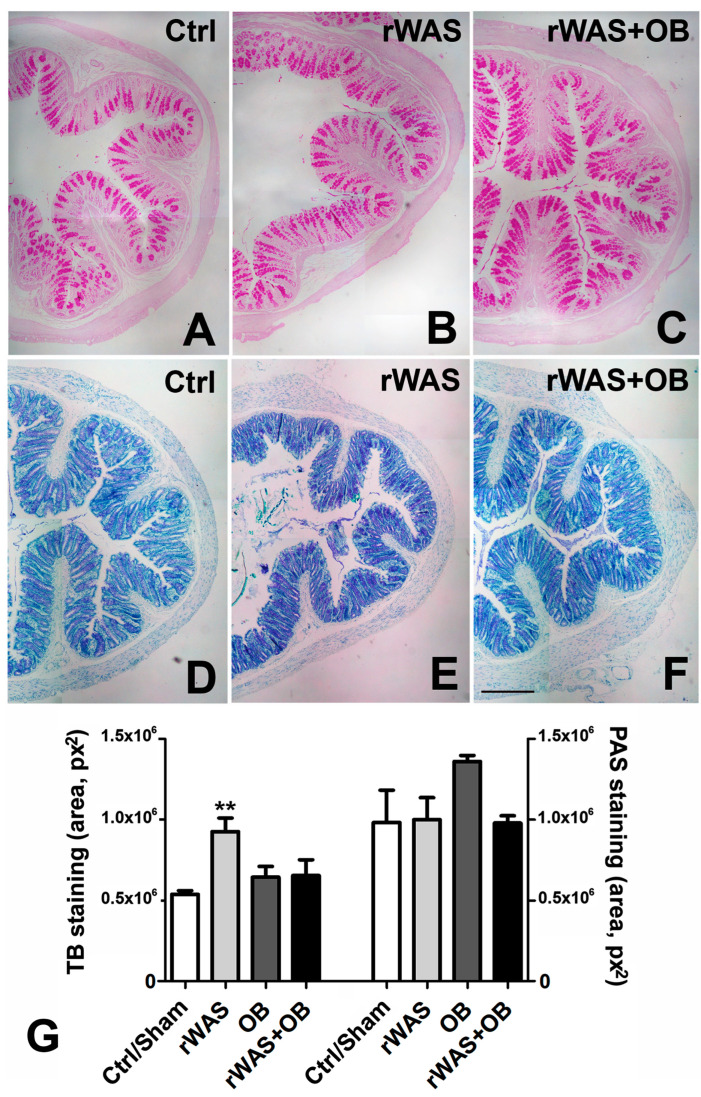
Mucus secretion. Periodic Acid and Schiff’s reagent (PAS) (**A**–**C**) and Toluidine Blue (TB) staining (**D**–**F**). The two stainings were located in the glandular funds and in the goblet cells. Quantitation of PAS staining showed no difference in the total quantity of mucus production among the groups of rats ((**G**), right columns). Conversely, the quantitation of the acidic component, measured as TB metachromasia, demonstrated a significant increase in the rWAS group compared with the other three ((**G**), left columns). ((**A**–**F**) bar = 50 µm). One-way ANOVA followed by Newman–Keuls post-test, ** *p* < 0.005. (n = 7), Bar = 100 μm.

**Figure 2 ijms-24-07440-f002:**
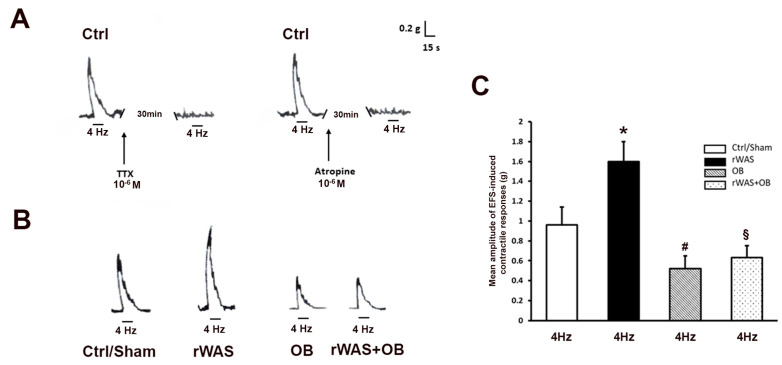
EFS in preparations from the different animal groups and effects of TTX and atropine. (**A**): Typical tracings showing the contractile response to EFS in preparations from Ctrl rats. The response elicited by EFS (4 Hz) is abolished following the addition to the bath medium of either TTX (left-hand panel) or atropine (right-hand panel). (**B**): Typical tracings showing the contractile response to EFS in preparations from: Ctrl, rWAS, OB, and rWAS+OB rats. The amplitude of the neurally induced contractile response to EFS is higher in preparations from rWAS in respect with that recorded from the Ctrl ones. In preparations from both OB and rWAS+OB rats, the amplitude of the contractile response to EFS appears greatly reduced in comparison with that of the rWAS or even Ctrl ones. (**C**): Bar chart showing the mean amplitude of the EFS-induced contractile responses in preparations from the different groups. No statistical differences were observed in the amplitude of the contractile responses to EFS between preparations from Ctrl and Sham rats, so all these data were put together and analyzed (Ctrl/Sham). The mean amplitude of the contractile responses to EFS in preparations from rWAS animals is increased in comparison with the Ctrl/Sham rats. At variance, that of both OB and rWAS+OB rats is reduced, without statistical differences between these two groups, in comparison with Ctrl/Sham and rWAS ones. All values are means ± S.E.M. of 8–12 preparations. * *p* < 0.05 vs. Ctrl/Sham; # *p* < 0.05 vs. Ctrl/Sham or rWAS; § *p* < 0.05 vs. Ctrl/Sham or rWAS, and *p* > 0.05 vs. OB (ANOVA and Newman–Keuls post-test).

**Figure 3 ijms-24-07440-f003:**
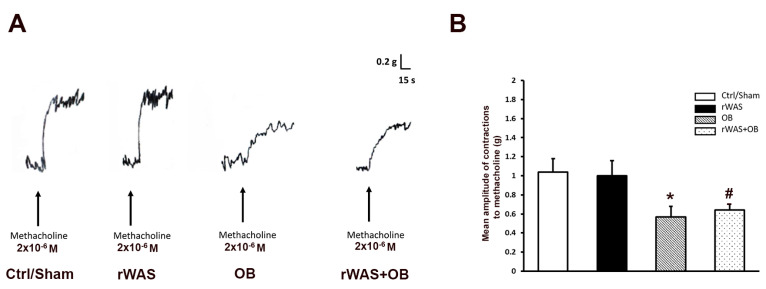
Direct muscular responses elicited by methacholine in strips from the different animal groups. (**A**): Typical tracing showing the muscular contraction elicited by the addition of methacholine to the bath medium in preparations from Ctrl, rWAS, OB, and rWAS+OB rats. Note the similar amplitude of the response in preparations from Ctrl and rWAS animals and its reduction in both OB and rWAS+OB rats. (**B**): Bar chart showing the mean amplitude of the direct contraction induced by methacholine in preparations from Ctrl/Sham, rWAS, OB, and rWAS+OB rats. The amplitude of the response to methacholine was not statistically different between Ctrl and Sham rats, so all the data were put together and analyzed (Ctrl/Sham)**.** No statistical differences are present in the mean amplitude of the response to methacholine between preparations from Ctrl/Sham and rWAS rats. At variance, the mean amplitude of the direct response to methacholine appears reduced in preparations from both OB and rWAS+OB rats, without statistical differences between these two groups, in comparison with the Ctrl/Sham and rWAS ones. All values are means ± S.E.M. of 6–8 preparations. * *p* < 0.05 vs. Ctrl/Sham or rWAS; # *p* < 0.05 vs. Ctrl/Sham or rWAS, and *p* > 0.05 vs. OB (ANOVA and Newman–Keuls post-test).

**Figure 4 ijms-24-07440-f004:**
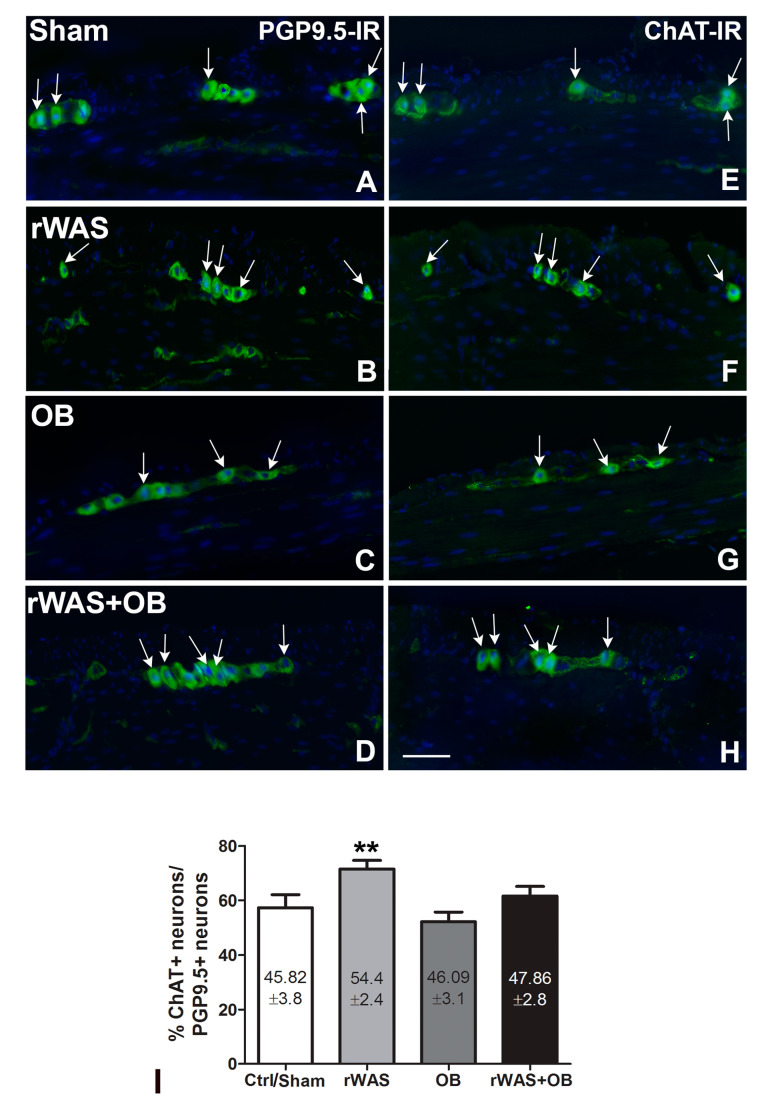
Protein Gene Product 9.5 (PGP9.5) and Choline Acetyl-Transferasi (ChAT)-immunoreactive (IR) myenteric neurons. PGP9.5 (**A**–**D**) and ChAT (**E**–**H**) labeling (green) in the myenteric neurons of Sham (**A**,**E**), rWAS (**B**,**F**), OB (**C**,**G**), and rWAS+OB (**D**,**H**) rat distal colon. Hoechst 33342 labeled the nuclei (blue). The arrows indicate the neurons that express both PGP9.5 and ChAT. Bar = 50 μm. Quantitative analysis of the percentage (%) of ChAT-IR neurons respect to the total PGP9.5-IR neurons and of the total number of ChAT-IR neurons. (**I**). The columns express the %. The number of ChAT-IR neurons is reported inside the columns. Both parameters showed a significant increase in the rWAS compared with all the other groups. Only the labelled neurons containing the nucleus were included in the statistical analysis. One-way ANOVA followed by Newman–Keuls post-test, ** *p* < 0.005. (n = 5).

## Data Availability

Not applicable.

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
