# Peer review of "Otilonium Bromide Prevents Cholinergic Changes in the Distal Colon Induced by Chronic Water Avoidance Stress, a Rat Model of Irritable Bowel Syndrome"

_ijms, 2023, doi:10.3390/ijms24087440_

Round 1

Reviewer 1 Report

Thank you for the review invitation. I enjoyed reading “Otilonium bromide prevents cholinergic changes in the distal colon induced by chronic water avoidance stress, a rat model of irritable bowel syndrome” by Traini et al. In this study an animal model for irritable bowel syndrome is used in rats. The authors compared the use of otilonium bromide to a control group in rats with repeated water avoidance stress. Increased chemical and contractile responses were seen, and were reduced by otilonium bromide administration. This is a relevant topic, as irritable bowel syndrome has a huge disease burden in human.

There are a few issues that need to be solved:

1. What was done with animals that were ill or died? Were they replaced?

2. How was the sample size achieved?

3. Section 4. 1 is not completely clear.

Why male rats?

How were they randomly divided?

May details are missing on handling of the animals. Perhaps a protocol should be included.

4. In the discussion or conclusion section, please make a suggestion for future research.

Reviewer 2 Report

Otilonium bromide (OB) is a medication used to treat irritable bowel syndrome (IBS). OB is a selective antispasmodic that acts directly on the colon, and it has the following pharmacological effects:

Antispasmodic effect: OB increases local blood flow in the colon and promotes muscle relaxation by inhibiting the release of acetylcholine (ACh), a neurotransmitter. This has the effect of relieving muscle spasms in the colon.

Antisecretory effect: OB inhibits mucus secretion in the colon, improving symptoms such as constipation.

Anti-inflammatory effect: OB has an anti-inflammatory effect in the colon.

Antibacterial effect: OB maintains the balance of bacteria in the colon by inhibiting the proliferation of bacteria.

Antimicrobial effect: OB has an effect similar to antibiotics against bacteria in the colon, inhibiting the growth of bacteria.

Therefore, since OB acts on various aspects of IBS symptoms and is used as a medication to improve IBS symptoms, it is recommended to provide sufficient descriptions of the research results known as previous studies on OB.
